# The Drivers of Mesozoic Neoselachian Success and Resilience

**DOI:** 10.3390/biology14020142

**Published:** 2025-01-30

**Authors:** Manuel Andreas Staggl, Carlos De Gracia, Faviel A. López-Romero, Sebastian Stumpf, Eduardo Villalobos-Segura, Michael J. Benton, Jürgen Kriwet

**Affiliations:** 1Department of Palaeontology, Faculty of Earth Sciences, Geography and Astronomy, https://ror.org/03prydq77University of Vienna, Josef-Holaubek-Platz 2, 1090 Vienna, Austria; 2Vienna Doctoral School of Ecology and Evolution (VDSEE), https://ror.org/03prydq77University of Vienna, Djerassiplatz 1, 1030 Vienna, Austria; 3Departamento de Zoología, Facultad de Ciencias Naturales, Exactas y Tecnología, https://ror.org/0070j0q91Universidad de Panamá, Panama 0824, Panama; 4https://ror.org/035jbxr46Smithsonian Tropical Research Institute, Balboa, Ancon, Panama P.O. Box 0843-03092, Panama; 5EvoDevo Research Group, Unidad de Sistemas Arrecifales, Instituto de Ciencias del Mar y Limnología, https://ror.org/01tmp8f25Universidad Nacional Autónoma de México, https://ror.org/01tmp8f25UNAM, Puerto Morelos 77580, Quintana Roo, Mexico; 6School of Earth Sciences, https://ror.org/0524sp257University of Bristol, Life Sciences Building, Tyndall Avenue, Bristol BS8 1TQ, UK

**Keywords:** environmental factors, evolution, sharks, rays, skates, diversity, climate change, detrended correspondence analysis

## Abstract

The modern diversity of sharks, skates, and rays (Neoselachii) is the result of various diversification and extinction events during the Mesozoic (252–66 Ma). However, the key drivers of their diversity patterns remain poorly understood despite all the progress that has been accomplished in recent years. Here, we show that the interplay of climatic- and tectonic-linked trajectories, resulting in a high shallow marine habitat availability and lower atmospheric CO_2_ concentration, were significant drivers and sustainers of Mesozoic neoselachian diversity. We show, for the first time, that higher atmospheric CO_2_ content negatively affected neoselachian diversity in the past. The recognized gradual faunal changes throughout the Mesozoic and the two major diversification events during the Jurassic and Cretaceous, respectively, ultimately cumulated in an all-time diversity high in the Palaeogene despite the events during the end-Cretaceous extinction event, highlighting their remarkable resilience and adaptability despite severe environmental challenges. We thus provide novel perspectives on the processes underlying neoselachian diversification since the Mesozoic that contribute importantly to a better understanding of the selective forces that have shaped the long-term evolution and diversification of neoselachians. Given their vital role in modern ecosystems, our results provide information about possible future trends in the face of the current climate crisis.

## Introduction

1

Neoselachii, i.e., modern sharks, rays, and skates [1], plus their extinct immediate relatives, including Synechodontiformes [2], have a long evolutionary history. Their fossil record, dating back to the Permian (~290 Ma), reveals the successive evolution of stem group members and extant lineages [3,4]. Neoselachian diversity during the Mesozoic was particularly affected by a series of diversification events that started in the Early Jurassic after the end-Triassic extinction event onwards, leading to their modern representation and composition [5-8] with an extant diversity of more than 1200 known species (Fricke et al. [9], accessed 29 August 2024). As neoselachians include keystone taxa, as well as abundant mesopredators within marine food webs, their conservation is essential to maintaining the functionality of the entire ecosystem [10,11]. Given the great variety of adaptations among Neoselachii, preserving the diversity of the whole clade is of even higher importance than protecting just individual species to prevent cascading effects from restructuring marine food webs [11,12]. Although the current climate crisis differs from previous climate changes, especially in terms of speed and magnitude, the examination of past responses of particular groups to environmental change is of immense importance [13,14]. Climatic shifts, changes in habitat availability, and biotic interactions were previously shown to drive levels of biodiversity [15]. Understanding the underlying processes that led to modern diversity and faunal composition of taxa—in this case, neoselachians—is a fundamental element in tracing a taxon’s evolution [14].

Here, we present a quantitative analysis of sampling-standardized diversity dynamics of neoselachians in deep time based on the largest genus-level dataset to date. This dataset comprises over 20,000 fossil occurrences from the Mesozoic and Cenozoic, encompassing more than 470 genera. The assembly of this dataset was conducted by incorporating a number of publicly accessible data sources (for further details, please refer to the “Section 2” section below). Performing a detrended correspondence analysis allowed us to identify changes in neoselachian faunal compositions through time. This represents the first application of this ordination method for investigating neoselachian faunal turnovers in deep time. By subsequently employing stepwise multivariate linear modeling, we present evidence of the influence of environmental changes on the diversity and faunal composition of Mesozoic neoselachians.

The results of the present study support previous research and suggest an episodic diversification history for neoselachians [6-8,16-19]. The analyses reveal the interplay between fundamental diversification drivers and the specific conditions that shaped Mesozoic neoselachian diversity and faunal composition changes. Our results emphasize the resilience and adaptability of neoselachians, with each major extinction event being succeeded by a notable recovery in diversity, which typically exceeds the diversity levels observed prior to the extinction event. Amongst others, we identified rising atmospheric CO_2_ concentrations, for the first time, as a substantial influencing factor in neoselachian diversity and faunal composition in the Mesozoic. This finding has tremendous importance for understanding the fate of neoselachians in the future.

## Materials and Methods

2

### Fossil Occurrence Data and Diversification Analyses

2.1

#### Fossil Occurrences

2.1.1

The primary data were assembled using the Paleobiology Database (paleobiodb.org; Downloaded from the PBDB on 19 June 2023, using the taxon name Elasmobranchii). These data were subsequently supplemented by the addition of fossil data from numerous scientific collections to increase samples of poorly represented regions, as recommended by Henderson et al. [20]. Data were collected via queries at GBIF [21] (accessed 11 September 2023), IdigBio [22] (accessed 26 June 2023), and additionally by queries on the website of fossil collections in natural history museums (e.g., Natural History Museum, Royal Belgian Institute of Natural Sciences). Additional entries in the database were made by adding occurrences found during an extensive literature search. We used genera instead of species to rule out uncertainties associated with species-level identifications due to fragmentary material [14,16,18,19,23,24]. Only specimens that included genus-level identification and stratigraphic information (layer, stage, or Ma) were considered. Dubious or poorly justified occurrences were excluded to avoid biases in diversity and/or first (FAD) and last appearances (LAD). All occurrences were checked for correct taxonomic assignment and validity, and in the case of outdated classifications, they were corrected to the latest accepted taxonomic assignment and systematic affiliation. The general taxonomy followed was that of Compagno [1], summarizing the modern shark lineages, batoids, and Synechodontiformes under the term Neoselachii. Other potential, especially Palaeozoic, neoselachian taxa were not considered because of their uncertain systematic status. The validity of each taxon (order, family, and genus) was checked using the bibliography database Shark References [25] (accessed between June 2023 and December 2023). The taxonomy was updated as and when necessary in order to reflect the most up-to-date knowledge on the subject. No subgenera were used (e.g., instead of *Otodus* (*Carcharocles*) sp., *Otodus* (*Otodus*) sp., or *Otodus* (*Megaselachus*) sp., *Otodus* sp. was used for all subtaxa).

#### Assignment of Occurrence Age

2.1.2

Each occurrence with a preassigned stage or geological formation was placed as accurately as possible using the latest geological timescale taken from the International Chronostratigraphic Chart v2023/09 [26]. Each occurrence was assigned a minimum and maximum age (e.g., upper and lower boundaries of the individual geological strata). For example, if a specimen was placed in the Albian without any additional information, we assigned a minimum age (FAD) of 113 Ma and a maximum age (LAD) of 100.5 Ma. In sum, this resulted in a range of occurrences, with the first appearance date (FAD) not necessarily being the exact time of origination of a taxon due to the fact that the true origination of the taxon could be beyond fossil occurrence. The same applies to last appearance dates (LADs) [27].

#### Final Fossil Datasets

2.1.3

We were able to assemble a dataset of 20.028 neoselachian fossil occurrences comprising 471 genera in 16 orders ranging from the Permian to the Holocene as the final database. Although the focus of this study is on Mesozoic diversity patterns, Cenozoic data were included to avoid edge effects at the KPg boundary. A table of all included genera with their respective FADs and LADs is included in the supplementary file (Supplementary Materials). We conducted a rarefaction analysis in R 4.3.2 [28] using the package vegan (v.2.6-4.) [29] and implementing a CI of 0.95 to evaluate the sampling coverage of the assembled dataset. This was performed for the dataset as a whole and for individual periods (Triassic, Jurassic + Cretaceous, Cenozoic) (Supplementary Materials).

#### Analyzing Evolutionary Dynamics

2.1.4

All diversity analyses and sampling standardizations were performed using the R package divDyn (v.0.8.2.) [30]. Plots were created using the package ggplot2 (v.3.5.0.) [31]. The dataset was divided into 58 bins of 6 Ma each and analyzed in these intervals to avoid bias regarding stage duration [15,20,32]. For that, a mid-point between the maximum and minimum ages of each occurrence was set and subsequently placed according to this mid-point in the respective time bin [20]. The diversity analyses were performed using the command “divDyn” for neoselachians as a whole and selachians and batoids separately. This provided the first results on diversity, which still contained sample biases. To minimize the latter, the data were subsampled using the shareholder quorum subsampling approach [32,33] in the package divDyn (v.0.8.2.) [30] with the function “subsample”. A quorum of 0.4 at 1000 iterations produced the most consistent diversity curves and was used accordingly for the final analyses. Based on these subsampled data, the respective origination and extinction rates were determined. Diversity was measured using three complementary approaches, sampled in bin (SIB), range through (RT), and boundary crosser (BC) [34].

#### Analyzing Faunal Composition

2.1.5

A detrended correspondence analysis (DCA) was performed in R 4.3.2 [28] using the function “decorana” in the package vegan (v.2.6-4.) [29] to obtain insights into the faunal composition of neoselachian diversity over time, following the method of Correa-Metrio et al. [35]. Detrended correspondence analysis can be used to track ecological changes through time, identify different ecological spaces through a priori interpretation of taxon ordination, and quantify ecological turnover among samples. These detrended correspondence analyses thus differ from other ordination methods, such as principal components analysis (PCA) and correspondence analysis (CA). First, it requires few prior assumptions, and the results can be directly interpreted in terms of ecological turnover, while PCA and CA have been shown to have rather unbalanced ordinations, which can distort the correct identification of environmental parameters and the conclusions drawn from the results [35-37]. This analysis was used previously in various studies to reconstruct past ecological changes, such as changes in vegetation and climate ([35] and the literature therein). The DCA showed that the highest relative importance lies mainly in DCA axis 1 (DCA1) followed by axis 2 (DCA2) (Supplementary Materials). Axes 1 and 2 were plotted together to display the position of individual faunas within each time bin (Supplementary Materials). We further used the results of the DCA1 as a dependent variable of the stepwise multivariate linear regression described below.

### Selection of Diversification Drivers

2.2

Several abiotic and biotic parameters were selected to be correlated with neoselachian diversity to test for possible diversity drivers through time. These datasets were selected based on their recency, completeness for the analyzed period, and tight binning. Linear interpolation was used to estimate data points between the two nearest points in time around the respective bin (Supplementary Materials). We favored datasets with narrower time bins for individual environmental parameters to achieve the smallest possible deviation in the values. Sea level data were obtained from Haq [38], sea surface temperature data were sourced from Song et al. [39], Mesozoic atmospheric CO_2_ concentration data were obtained from Foster et al. [40], flooded continental area data came from Marcilly et al. [41], and fragmentation data were sourced from Zaffos et al. [42].

Possible biotic drivers of neoselachian diversity include dinoflagellate diversity (MacRae, unpublished data, as indicated by Katz et al. [43]), calcareous nannoplankton diversity [44], planktic foraminifera diversity [45], and bony fish diversity (paleobiodb.org; Downloaded from the PBDB on 19 August 2023, using the taxon name Actinopterygii). The use of diversity as an indicator of the abundance of primary producers (and bony fishes) and productivity is legitimate, although it may not be completely precise [46]. Previous studies were conducted based on this assumption, which follows the principle that speciation rates change with environmental changes and that these changes also affect abundance in a similar way [47].

### Estimating Palaeoenvironment-Dependent Diversification

2.3

Spearman’s rank correlation analyses were initially performed to determine the environmental parameter that showed the strongest general correlation with neoselachian diversity or faunal composition (DCA1). Subsequently, stepwise multivariate linear regression and the Akaike information criterion (AIC) were used to assess the relative likelihood of a number of *a priori* models, following the method used by Marx and Uhen [46]. The analyses were performed in R 4.3.2 [28] using the functions “lm” and “step”. The dependent variables chosen for analysis were neoselachian-, selachian-, and batoid diversity, based on the results from the SQS and the results for axis 1 of the DCA.

## Results and Discussion

3

### Diversification Patterns

3.1

Modern neoselachians emerged in a stepwise fashion through the past 200 Ma, with some apparent acceleration in origination rates from the Late Cretaceous to Paleogene, between 100 and 50 Ma (Figure 1, Figure 2, and Supplementary Materials). Our findings confirm earlier propositions in some respects. For example, it was previously suggested that extinction events are usually followed by higher diversification rates [15,48,49], and this also seems to apply to neoselachians, as each extinction event resulted in a subsequent diversification, which in turn resulted in a higher diversity than prior to the extinction (Figure 2). This hypothesis is confirmed in the more detailed diversity-through-time plots as range-through (RT), boundary-crosser (BC), and sampled-in-bin (SIB) diversities (Figure 2).

Triassic neoselachian diversity dynamics are notably volatile and much more unstable than in the subsequent periods. The low number of Triassic taxa and the initially rather fluctuating turnover rates suggested by the detrended correspondence analysis (DCA) are most likely caused by the small sample size and weak taxonomic resolution due to a relatively small number of taxonomic studies, resulting in high turnover rates even for small changes in the faunal community [7,8,50]. Overall, the DCA shows a reduction in the number of key contributing taxa back in geological time (Figure 3).

The Permian neoselachian fauna comprised exclusively stem group members, here synechodontiform genera, as no other unambiguous stem groups could be identified (Figure 3). Due to the first appearance of new taxa (genera) of unknown affinities, the synechodontiform proportion of the total neoselachian fauna was reduced to 50% (Supplementary Materials, bin 20 vs. bin 22). This dramatic reduction coincides with the timing of the Manicouagan asteroid impact (~214 Ma) [51] and marks a turning point in faunal composition toward lower DCA values. This impact was formerly identified as the cause of the end-Triassic extinction event [52]. However, it occurred 15 Ma before the Triassic–Jurassic boundary [51], making the impact unlikely as the underlying cause, although Onoue et al. [53] linked it to the extinction of at least some larger marine groups. The extent to which this shift in faunal composition change can be associated with an impact is thus questionable. It is more likely caused by the noted first appearance of numerous new neoselachian taxa in the fossil record that might have been better adapted and thus outcompeted ancient groups. The end-Triassic extinction slowed the faunal turnover, somewhat coinciding with the appearance of the first crown group of neoselachians [54,55] in the earliest Jurassic (Figure 1).

The Jurassic saw the initial marked diversifications of neoselachians, with the emergence of numerous orders that have persisted until the present day. The last evident shift in faunal composition change before the Cenozoic can be dated to approximately 183 Ma, at the transition from the Pliensbachian to the Toarcian, when numerous new groups appear in the fossil record, such as Rajiformes [55], Apolithabatiformes [56], Squaliformes [57], Rhinopristiformes [58], Orectolobiformes [59], Lamniformes [60], and Heterodontiformes [57]. The other six neoselachian orders known today appeared successively in the fossil record, rather than in clusters or simultaneously (Figure 1).

The Early Jurassic radiation was stated to be facilitated by rapid colonization of vacant niches following the end-Triassic extinction [8], the increase in shallow epicontinental seas through the Rhaetian Transgression with the subsequent breakup of Pangaea [5,7], and the emergence of new morphological adaptations due to the ability to readily modify general body plans and tooth morphologies [5,7,50,61]. The relatively constant diversity prior to the Jurassic/Cretaceous (J-C) boundary (Figure 2) can be explained as reflecting relatively stable conditions and that the prevailing background diversification is due to local and regional events that forced the respective faunas to evolve [50]. In contrast to earlier studies [7,8,18,50] that identified the Jurassic peak of diversity in the late Early or Middle Jurassic, respectively, our sampling standardized results indicate that the highest Jurassic diversity occurred in the Tithonian. The notable decrease in diversity toward the end of the Jurassic cannot be attributed to an edge effect at the J-C transition as assumed previously [50] as our analysis was carried out across the boundary, and this decrease therefore reflects a rather genuine pattern. Exploring the origination (Ori, Figure 2) and extinction (Ext, Figure 2b) rates in detail reveals that the Early Jurassic rise in diversity (Figure 2) occurs with high origination (Figure 2) and simultaneous low and stable extinction rates (Figure 2). Diversity continues to rise through the Middle and Late Jurassic, with fluctuating origination and extinction rates.

The Cretaceous experienced the most pronounced radiation in the evolutionary history of neoselachians so far, leading to the diversification of lamniform sharks especially and, to a lesser extent, other groups such as Orectolobiformes. In the Late Jurassic and Early Cretaceous, relatively constant faunal changes were observed (Figure 3). During this period, modern orders that were still absent from the neoselachian assemblage emerged gradually (Figure 1). This coincides with the initial breakup of Pangaea, which gave rise to the North and South Atlantic and Indian oceans [62]. The relative proportion of the modern orders as part of the full neoselachian diversity remained remarkably constant from the end of the Jurassic to the present day (Figure 3 and Supplementary Materials). The sharp drop in origination rates and rise in extinction rates at the J-C boundary (145 Ma), as described earlier [50,63], could reflect the occurrence of an extinction event triggered by environmental perturbations [64] or simply is an artifact of incomplete sampling [64] (Figure 2). No important neoselachian clades went extinct around the J-C boundary, and Early Cretaceous diversity was much higher than in the Late Jurassic [7,18,50], although few occurrences are typically reported from the Early Cretaceous [7,8,64]. The Jurassic neoselachian radiation proceeded relatively steadily across the J-C boundary and lasted until the middle Albian (107 Ma) (Figure 2). A sharp increase in origination rates started in the Berriasian (143 Ma) and peaked in the Hauterivian (131 Ma), while extinction rates remained lower than the origination rates and relatively stable throughout this time (Figure 2).

Guinot and Cavin [8] and Condamine et al. [65] reported a strong diversification of lamniform sharks in the Early Cretaceous, which also is supported by our analyses. This radiation persisted until the middle Albian but subsequently depicted a massive decline based on RT and SIB data (Figure 2 and Supplementary Materials). This, however, is not reflected in their extinction rates (Figure 2) and was already reported in the past to possibly reflect uneven sampling [7]. Surface sampling was the most common collection method, where the large, more robust, and distinctive lamniform remains were gathered more extensively and smaller taxa such as batoids, squaliforms, or carcharhiniforms were rarely found and reported [7,66,67]. This results, together with taphonomic biases favoring larger and more robust teeth, in a pronounced dominance of lamniform fossil occurrences, which significantly distorts the observed overall diversity patterns for neoselachians, even when subsampling is employed [67].

The second most diverse group in raw diversity (Supplementary Materials), the Orectolobiformes, diversified in the Early Cretaceous, especially in the earliest part, despite an extinction phase coinciding with the J-C boundary. As previously mentioned, the observed origin of this clade is dated to the Early Jurassic (Figure 1). However, the estimated divergence from other galeomorph lineages is estimated to have occurred around the T-J boundary [59]. The sampling-standardized orectolobiform diversity indicates a collection gap in the middle Early Cretaceous (Supplementary Materials). Following this gap, at the transition to the Late Cretaceous, orectolobiform diversity increased significantly into the early Late Cretaceous. From then until the Late Eocene was a time of general stability for this group, with high diversities, despite minor fluctuations, even across the KPg boundary (Supplementary Materials).

Cretaceous batoid diversity shows an increase from the mid-Early Cretaceous until the Late Cretaceous boundary, although this can hardly be described as a significant radiation event [7]. Only a few batoid families were described from the Early Cretaceous, including members of Rhinobatidae, Spathobatidae, and Sclerorhynchidae and one species of Dasyatidae (*Dasyatis speetonensis* Underwood et al. [68], an assignment that is rather dubious). Diversity increases steeply from the Aptian (125 Ma), especially at the transition to the Late Cretaceous (100.5 Ma). The middle Late Cretaceous saw a severe decrease in batoid diversity that cannot be clearly linked to any specific event [69] (Supplementary Materials) and can be likely attributed to a sampling bias [69,70]. The massive decline at the Early to Late Cretaceous boundary is followed by the most striking diversification in neoselachian evolutionary history to that point. Taxa within Lamniformes, Rajiformes, Orectolobiformes, and Carcharhiniformes all contribute to the significant peak in late Maastrichtian (66 Ma) diversity (Figure 2 and Supplementary Materials).

The KPg boundary event (66 Ma) caused a notable decline in neoselachian diversity, which nevertheless attained its highest diversity so far in the Middle Eocene. Evidence of the post-KPg decline is apparent in the SIB, but less pronounced in the RT diversity curves (Figure 2). Origination declined significantly thereafter and reached its minimum in the middle Ypresian (53 Ma) (Figure 2). The considerable post-KPg decline recuperated within a 21–25 Ma time interval. This recovery is somewhat accelerated amongst others by the first appearance of a considerable number of new batoid taxa in the fossil record shortly before the KPg boundary event and in the earliest Neogene [71] (Supplementary Materials). These contributed, amongst others, to a new peak in diversity in the Late Eocene (41.2 Ma) (Figure 2). Despite a modest declining trend in lamniform diversity up to the Late Eocene, this order remains among the most diverse groups within neoselachians. Previously supposed competition with Carcharhiniformes resulting in a decline of lamniform sharks [65] cannot be deduced from the subsampled data for the Late Cretaceous, with diversity patterns of both orders in the Neogene being similarly high and following similar trends (Supplementary Materials and Supplementary Materials). The Cenozoic peak of lamniform diversity is in the Middle to early Late Eocene, whereas carcharhiniform diversity peaked only later, at the Eocene-Oligocene transition. This diversity peak represents the all-time high of carcharhiniform diversity throughout their evolutionary history, as their diversity subsequently declines in both raw (Supplementary Materials) and sample standardized data (Supplementary Materials). In contrast to our results, Condamine et al. [65] found some competition between the two groups related to medium and large representatives, only after the post-KPg recovery. They also concluded that this might be due to passive exchange rather than direct replacement. Oligocene neoselachian diversity shows a significant drop in all three metrics (Figure 2), lasting until the end of the Neogene (2.6 Ma). Origination rates also show a pronounced decline already starting in the Bartonian (41.2 Ma) and lasting until the end of the Neogene (2.6 Ma). Extinction rates show a similar decline after the Priabonian (35 Ma) peak, reaching a minimum in the Rupelian (29 Ma) before increasing significantly until the end of the late Miocene (Figure 2).

The most striking change in neoselachian fauna since the Triassic occurs during the Oligocene–Miocene climatic transition. This transition is generally considered to represent a temporary global cooling event with an expansion of Antarctic ice sheets [72]. Further detailed investigation of this period with smaller binning doing justice to the different durations of epochs will have the potential to elucidate the specific causes of the most significant shift in faunal composition since the Triassic in more detail (Figure 3).

### Environmental Diversification Drivers

3.2

In order to better understand the drivers and sustainers of neoselachian diversity over time, we compared possible abiotic environmental drivers, i.e., sea level (SL), sea surface temperature (SST), atmospheric CO_2_ concentration (CO_2_), flooded continental area (Flood), and continental fragmentation (FragInd), as well as biotic drivers, i.e., dinoflagellate (Dino), calcareous nannoplankton (Nanno), planktic foraminifera (Foram), and bony fish diversities (Bony), that might have impacted neoselachian diversities in deep time (see Section 2 for detailed references of each parameter).

#### Abiotic Drivers

3.2.1

Our analysis reveals that the diversity indices derived from the individual diversity approaches exhibited varying outcomes when correlated with abiotic extrinsic factors. Accordingly, sampled-in-bin diversity (SIB) correlates generally weakly or insignificantly with neoselachians (sharks, rays, and skates) and selachians (sharks), respectively, with all tested abiotic parameters (Supplementary Materials). In contrast, SIB for batoids (rays and skates) showed significant and positive correlations with several abiotic drivers (see below). The correlations for all three subgroups for ranged-through (RT) and boundary-crosser (BC) diversities were overall strong and significant for most environmental drivers, except for SST. All three taxonomic groups, neoselachians as a whole, and selachians and batoids, respectively, display the strongest correlations with the fragmentation index and sea levels. Significant correlations were found between DCA axis 1 (DCA1) and all abiotic predictors, except for SST. All correlations, except for atmospheric CO_2_, were negative, indicating a shift in DCA1 values (Supplementary Materials).

The results of the stepwise multivariate linear modeling show how different results for the individual metrics of diversity through time (Supplementary Materials). Given the inherent volatility of SIB curves for neoselachians and selachians, respectively, these models tend to demonstrate comparatively diminished explanatory power relative to approaches that yield more stable curves as their adjusted R-squared was considerably lower than for the other diversity approach models. The batoid SIB models achieved a good adjusted R-squared, as the SIB diversity for this group showed greater stability than for the majority of other investigated subgroups and mirrored the dynamics observed in the RT diversity closely.

The BC, RT, and DCA1 models consistently produced very high adjusted R-squared values for all groups (Supplementary Materials). BC models generally showed stronger and more significant associations between abiotic factors and diversity compared to SIB models. RT models exhibited similar residual standard errors as SIB models, but with considerably higher F-statistic values, again suggesting stronger explanatory power for RT compared to SIB. The DCA1 model had a low residual standard error and a high F-statistic value, indicating the significant impact of environmental drivers on faunal composition changes (Supplementary Materials). We found poor model fits for neoselachian and selachian SIB data, while batoid SIB data yielded a good fit (Supplementary Materials). The BC, RT, and DCA1 models generally exhibit better fits, with good to excellent adjusted R-squared values across all taxonomic groups. We identified the atmospheric CO_2_ concentration, flooded continental area, and fragmentation index as the most consistent and reliable predictors for the BC, RT, and DCA1 approaches and taxonomic groups (neoselachians, selachians, and batoids) (Figure 4, Supplementary Materials). Continental fragmentation actually appears in all models as the strongest predictor driving neoselachian diversity.

The potential role of the fragmentation index as a diversification driver in the marine realm has been previously discussed [17,42,73,74]. Valentine and Moores [75] saw the reason for the significance of continental fragmentation primarily in the enlargement of shallow-water areas due to an increase in coastline, habitat heterogeneities, spatial separation, and associated developing endemism. By creating natural geographical barriers, a higher fragmentation index promotes allopatric speciation by restricting the distribution of species with a low dispersal capacity [42,74]. Bush and Payne [15] discussed that the frequently raised problem of the apparent increase in diversity with increasing shallow marine habitat area is simply due to the relationship between the available area and the correlated increased preservation potential of fossils. They concluded, as did Guinot and Cavin [17], that this is unlikely to be true, indicating that it is rather a genuine signal. The emergence of new taxa is thus associated with a gradual increase in habitats due to continental flooding corresponding to the species–area relationship, while extinction follows habitat depletion [15,17]. Guinot [76] described that heterogeneous habitats are more likely to be more species-rich as they facilitate the finding and occupation of new niches for taxa. More homogeneous habitats, conversely, lead to a reduction in diversity and the dominance of few taxa.

Shallow-water areas are usually warmer than deeper marine regions and have higher levels of solar radiation, making shallow marine environments so attractive for numerous marine taxa [77]. These areas can be considered high-energy environments, facilitating higher mutation rates and shorter generation times [8,17,78] and supporting increased speciation [78]. In combination with possible upwellings on continental shelves and the associated nutrient input, these regions also enable increased primary production [79,80]. These, in turn, form the basis for food webs and enable nutrient transfer to higher trophic levels, with more abundant lower trophic levels enabling more energy to be transported to higher levels and more complex food webs to develop [81]. This enables/requires niching, i.e., specialization, which subsequently enables the emergence of new taxa. It should be noted, however, that this cannot be generalized for all taxa among Neoselachii, as individual groups appear to have specialized early to a variety of habitats, most likely following a rather “opportunistic” radiation scenario [50]. The availability of shallow marine habitats as an important diversification driver for individual clades within Neoselachii was also previously described for batoids, many of which seemingly favor shallow-water areas [82]. Guinot and Cavin [8] further described this pattern for members of Lamniformes, Carcharhiniformes, and Orectolobiformes. Considering the latter two orders in particular, even today, we see many representatives of important carcharhiniform families, and especially orectolobiforms, being mainly adapted to shallow marine areas [83]. In contrast to that, squalomorph sharks already exhibited a tendency to prefer cooler environments in the early stages of their evolutionary history [7,76]. These environments could be in higher latitudes, especially boreal regions and beyond, or they colonized deeper marine regions [7,76].

For the first time, our analysis reveals that atmospheric CO_2_ concentration is a significant neoselachian diversification driver in the Mesozoic. Nowadays, CO_2_ is often discussed as a greenhouse gas that contributes to global warming [84]. In the past, significantly elevated CO_2_ levels in the atmosphere have also led to much warmer conditions, allowing the expansion of warmer regions to higher latitudes [85]. However, the relationship between CO_2_ and diversity in the Mesozoic is reciprocal. Increased atmospheric CO_2_ levels result in increasing ocean acidification [86]. The current state of knowledge of the effects of ocean acidification on the behavior and physiological responses of sharks, including the effects on their fitness and resilience, was summarized by Zemah-Shamir et al. [87]. Essentially, modern neoselachians appear to be more tolerant to acidification than bony fishes to some extent due to their ability to regulate the acid–base balance through ion transport mechanisms in their gills and kidneys. The regulation of the acid–base balance in the body is accompanied by metabolic costs. Ocean acidification thus has a direct physiological effect on neoselachians, in addition to the metabolic costs associated with acid–base regulation. These effects include alterations in blood chemistry and metabolism, as well as changes in sensory and behavioral responses. Sensory and behavioral responses include, amongst others, olfactory and auditory stimuli, which can affect foraging and predator avoidance. Zemah-Shamir et al. [87] stated that the vast majority of studies they reviewed found a negative impact of elevated CO_2_ levels on sharks. Di Santo [88] additionally reported that acidification also alters the calcification of the cartilage skeleton in batoids whereby in some skeletal regions, the calcification may be reduced, while in others, it is increased, providing ambiguous signals of the effects. Nevertheless, the long-term effects of higher CO_2_ levels on neoselachians in particular appear to have been investigated little or not at all yet. In vivo studies conducted over multiple generations could provide better insights into the physiological effects of increased CO_2_ levels and ocean acidification on extant neoselachians. By demonstrating the inverse relationship observed between the atmospheric CO_2_ concentration and neoselachian diversity, we present, for the first time, to the best of our knowledge, evidence of past negative impacts of ocean acidification on neoselachians.

The absence of the sea surface temperature as an important influencing factor of neoselachian diversity could be explained by the fact that, in particular, smaller and nearcoastal neoselachians have a relatively high thermal tolerance [89]. In the context of suboptimal temperature conditions, a shift in distribution could occur, either to higher latitudes or deeper-water regions [90,91]. As previously identified for squalomorph sharks [7,76], this could also apply to other neoselachian groups in a similar way. This temperature-avoidance behavior becomes problematic when geographical barriers prevent migration to cooler regions, as can be seen today in semi-enclosed oceans such as the Mediterranean Sea [91]. The colonization of deeper-water areas is particularly problematic to describe in the fossil record, as relatively few truly deep-sea deposits with associated fauna are known [8,23].

Our findings suggest that sea level may not be a primary diversification driver. This may be explained by the fact that the absolute ocean depth does not appear to exert a notable influence on neoselachian diversity. The indirect effects of sea level, such as the extent of flooded areas, on the other hand, could be of greater significance. Nevertheless, it would be inaccurate to equate the sea level with the extent of the flooded area. The latter is additionally influenced by a number of further factors, including the regional geomorphology of the coastlines and the starting point of the sea level during a specific time [92].

#### Biotic Drivers

3.3.2

Once more, our results for the optimal biotic model, explaining the patterns of diversity among neoselachians, yielded diverging outcomes regarding the indices employed for the assessment of individual diversity. SIB models again achieved rather limited explanatory power with biotic drivers (again, except for batoids), whereas analyses of BC, RT, and DCA1 revealed significant relationships between neoselachian diversity and various biotic factors (Supplementary Materials). Notably, foraminiferan and bony fish diversities emerged as the two key drivers across different diversity metrics and taxonomic groups. The distinction between abiotic and biotic diversification drivers is necessary as it is possible that other taxonomic groups, like bony fishes, for example, may exhibit analogous responses to environmental influences as neoselachians. For instance, it has been shown that, in general, comparable diversification patterns can be observed for ray-finned fishes and neoselachians during the Mesozoic [16]. This makes a significant correlation between the diversification patterns of both groups difficult, insofar as it is either a matter of convergence or actual coevolution. While considerable radiation for neoselachians is observed in the Mesozoic, by far the most significant part of the radiation of ray-finned fishes took place in the Cenozoic [19,93,94]. Especially at the order and family levels, a large part of the neoselachian diversity had already appeared before the Cenozoic [19]. It can be reasonably argued that neoselachians, which tend to occupy higher trophic levels, are dependent on the underlying levels as part of bottom-up processes. Nevertheless, it is well known that food webs are characterized by complex dependency relationships. Consequently, the modeling of modern ecosystems represents a considerable challenge in itself [95], with the modeling of past food webs based on the fossil record representing an even greater challenge.

## Conclusions

4

General patterns: Our results confirm the two previously identified radiation events of neoselachians in the Jurassic and Cretaceous, respectively, and provide evidence of sampling biases at the transition from the Early to Late Cretaceous due to the dominance of lamniform sharks in terms of diversity and occurrence counts, which might be related to sampling techniques rather than being a genuine signal. Our data highlight the Jurassic as a key time in the diversification of neoselachians, especially at the order level, whereas the Cretaceous is of great importance at the family, and particularly the genus, levels. The KPg boundary event appears to be less severe for neoselachian diversity in terms of extinctions than previously assumed [6,8,65,96], although the recovery after the end-Cretaceous extinction event took a comparatively long time (21–25 Ma) but ultimately led to an all-time high in diversity during the Cenozoic.

Generalist lifestyle and resilience: Our results give clear evidence of the common theory that taxa that are able to adapt flexibly to a wide range of ecological niches are much better equipped to cope with changing conditions than very specialized groups that are only adapted to a narrow range of ecologies [15,97] (Supplementary Materials). These adaptations can include a wide geographical distribution [65,96,98], a relatively generalist lifestyle [96,99,100], or the ability to adapt to a wide variety of habitats and rapidly changing environmental conditions by altering the body plan or physiology [7]. In addition to the tendency to occupy higher trophic levels than other species within the same ecosystem [101,102], the capacity of neoselachians to recover from extreme incidents such as extinction events, and even emerge stronger in terms of diversity in the aftermath, demonstrates an impressive degree of evolutionary success coupled with remarkable resilience. The findings presented here provide quantitative evidence to support this commonly reported pattern (Figure 2 and Figure 3). Chin et al. [103] analyzed the extinction risk for neoselachians in today’s Great Barrier Reef. They concluded that the highest risk lies with individual highly specialized taxa. A generalist lifestyle has been identified as one of the key factors that contribute to the evolutionary success of neoselachians, with specialization often associated with higher turnover rates and a shorter median lifespan of a taxon [15,65,96]. When considering extant sharks, rays, and skates, a certain degree of specialization in some groups in terms of habitat, diet, etc., is discernible, but in general, they can be considered to be generalists. The slow and gradual turnover identified throughout most parts from the Jurassic until the Oligo-Miocene transition, without major alterations (Figure 3), provides further evidence of neoselachian resilience and adaptability to environmental changes, even during periods of substantial environmental changes such as the KPg boundary event.

Key drivers: The importance of the availability of heterogeneous habitats and shallow-water areas is highlighted by the strong dependency of neoselachian diversity on continental fragmentation, flooded continental areas, and sea surface temperature (Figure 4, Supplementary Materials). For the first time, we demonstrate the negative effect of higher atmospheric CO_2_ concentrations on neoselachian diversity in the past (Figure 4, Supplementary Materials). These findings encourage further studies on living sharks and batoids over prolonged time periods, aiming to reveal the direct effect of higher CO_2_ concentrations and the subsequent ocean acidification over multiple generations to evaluate the physiological effects. This may facilitate a more profound examination of the specific factors that contribute to the fluctuations in deep-time biodiversity associated with changes in CO_2_ levels. Of particular interest is the question of whether direct physiological effects or indirect consequences of CO_2_ on components within marine ecosystems are responsible for the observed changes in neoselachian diversity.

Importance of deep time diversity studies: Analyzing the effects of environmental changes on the diversity and composition of past neoselachian faunas has the potential to provide a better understanding of the response of extant sharks, rays, and skates to climate change. A recent study [104] showed that extant chondrichthyans (neoselachians and holocephalians) are among the most endangered taxa. Anthropogenic stressors such as overfishing and climate change are leading to a dramatic decline in populations, and extinction rates are reaching alarming levels. Studying the past is therefore crucial to understanding the impact of climate change and other environmental stressors on today’s biodiversity and developing effective conservation measures. Detailed investigations on the drastic decrease in diversity in the Cenozoic and the major shift in neoselachian faunal composition observed here at the Oligo–Miocene transition could provide novel and important information about these processes in the future. Nonetheless, an essential point to emphasize is that the outcomes of modeling should be regarded solely as approximations of actual phenomena, thereby offering insights into potential relationships that enable the establishment of hypotheses.

Our results derived from analyzing diversity patterns during the Mesozoic (Figure 4, Supplementary Materials) lend support to the hypothesis that in the absence of the three major drivers of ocean degradation (overfishing, pollution, and the anthropogenic rise in CO_2_ concentration [105]), extant neoselachians may potentially emerge as beneficiaries of the current climate crisis. An increase in shallow marine habitats due to an increase in flooded continental areas and warm conditions was already favorable for neoselachian diversification in the past and, thus, might also be in the future.

## Supplementary Material

1

## Figures and Tables

**Figure 1 F1:**
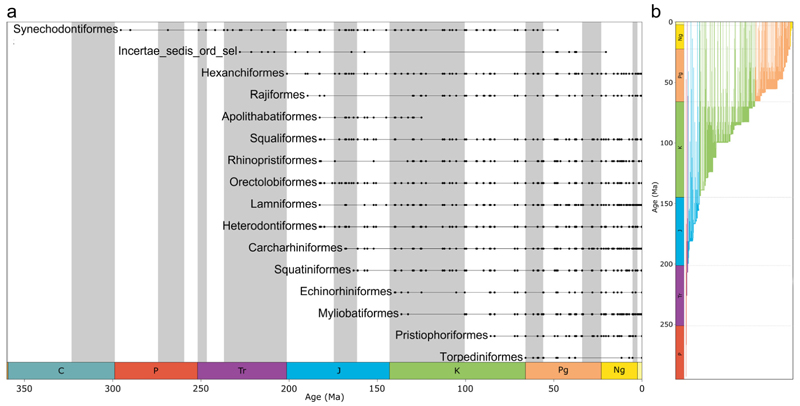
Chronostratigraphic range chart of neoselachian fossil occurrences: (**a**) Neoselachian order range through time. Solid black dots represent one or multiple fossil occurrences at the respective time. Solid black bars represent the entire period between first and last known occurrences of the individual taxon. (**b**) Neoselachian genus ranges through time. Solid bars represent the entire period between first and last known occurrences of the individual genus, colored according to the color of the stratigraphic period of the first known occurrence.

**Figure 2 F2:**
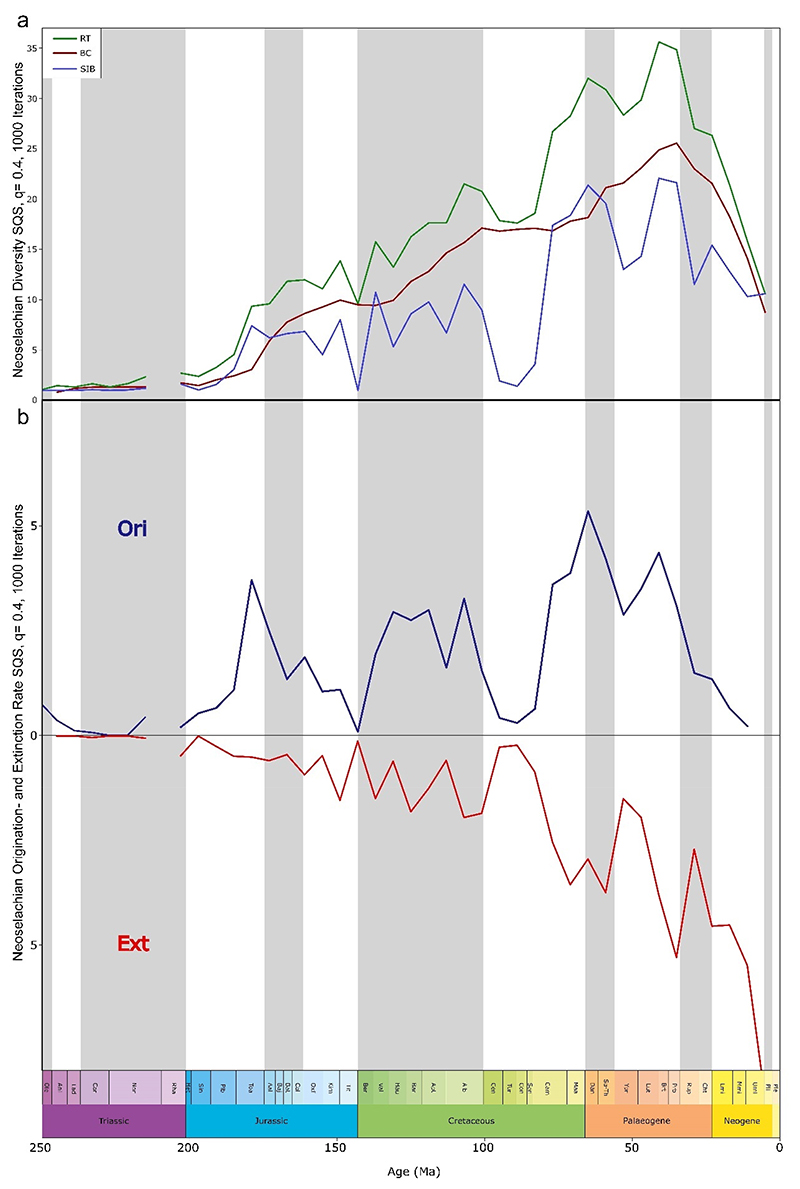
Diversity dynamics of neoselachians genera through the Mesozoic and Cenozoic based on shareholder quorum subsampled data (SQS, q = 0.4, 1000 iterations). (**a**) Standing neoselachian diversity. Diversity is displayed using three approaches, sampled in bin (SIB, blue solid line), range through (RT, green solid line), and boundary crosser (BC, red solid line). (**b**) Origination and extinction rate. Both diversity metrics are based on the subsampled data. Both parameters are displayed mirrored against each other via the x-axis. Distinct radiations can be observed in the late Early Jurassic and the mid-Early and mid-Late Cretaceous. The end Triassic discontinuity is a result of the sample standardization (uneven sampling at that time).

**Figure 3 F3:**
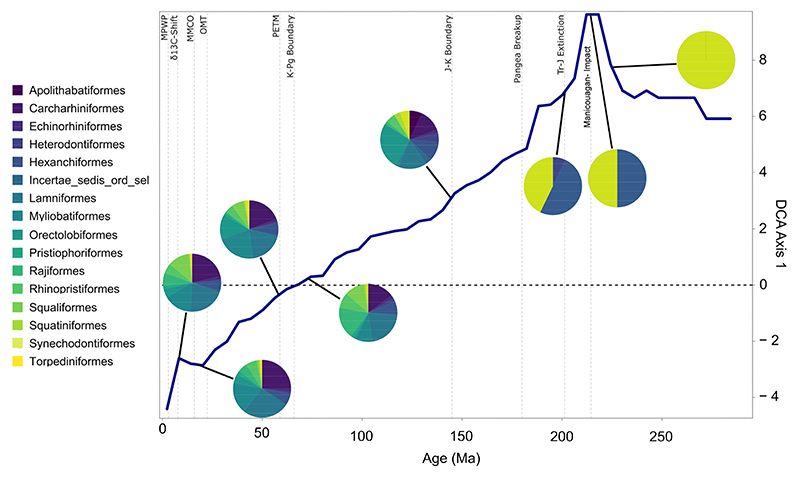
Rate of neoselachian faunal composition change shown by scores of samples on DCA axis 1 through time. Blue solid line represents the rate of faunal change in terms of their ordination resulting from the DCA rescaled faunal composition. Units of the y-axis are standard deviations as a metric for faunal turnover. Distinctive points through Earth’s history are indicated as references by the dashed lines. Pie charts show the faunal composition of the indicated time in terms of proportional genus richness of the respective orders. Abbreviations: MMCO, mid Miocene climate optimum; MPWP, mid Pliocene warm period; PETM, Paleocene–Eocene thermal maximum; OMT, Oligocene Miocene climatic transition.

**Figure 4 F4:**
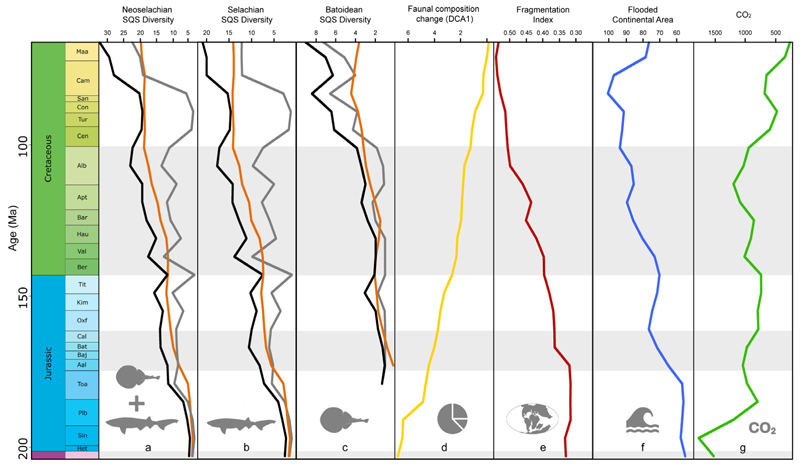
Comparison of neoselachian (**a**), selachian (**b**), and batoidean (**c**) genus diversity and neoselachian faunal composition change (**d**) with continental fragmentation (**e**), flooded continental area (**f**), and atmospheric CO_2_ concentration (**g**) through time. Diversity in (**a**–**c**) is shown as subsampled (SQS) sampled-in-bin (grey), boundary-crosser (orange), and range-through (black) diversity. Data of (**e**–**g**) retrieved from published literature, respective reference mentioned in the material and method section. For full statistical results see Supplementary Materials.

## Data Availability

The data supporting the findings are available in the supplementary material. The taxonomic occurrence dataset is available from the corresponding author upon reasonable request. R programming codes are based on previously published methods but are available upon reasonable request to the corresponding author.

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
