# Peer review of "The Drivers of Mesozoic Neoselachian Success and Resilience"

_biology, 2025, doi:10.3390/biology14020142_

Round 1

Reviewer 1 Report

Comments and Suggestions for Authors

I must begin saying that complex numerical modellings were challenging during this current review. I do have concerns with relaying entirely on these type of modellings for assuring global hypotheses. Databases should be considered as an incomplete approaching instead of confident sources of data. This because many localities around the planet are undersampled and all of them has biases that will affect the results.

I do think that the structure of the manuscript and the carried out analysis is very valuable, but I would recommend to restrict it to localities with good sampling (i.e., large number of pieces) as a first goal, instead of pointing to global conclusions (which might be precipitate at this moment). In general, several sections (e.g., Environmental Diversification Drivers, Biotic Drivers) deal easily with planetary variables (sea level, sea surface temperature, atmospheric CO2 concentration , flooded continental áreas, dinoflagellate, nannoplankton, planktic foraminifera, etc) which in many cases are far from being accurately controlled in several localities with relevant chondrichthyan records. In addition, I really miss pivotal references on these sections, because each parameter has many previous studies and they are not reflected in the text. Again, my recommendation it that authors should build a solid case only restricted to those localities where these drivers were enough studied previously (adding to the previous recommendation of localities with good sampling numbers). Else, they risk to have artifactual results.

Regarding the Conclusions, these reinforce previous hypotheses on the Jurassic and Cretaceous radiation events, which is great. Regarding the lamniform dominance starting the Late Cretaceous, the authors point as a possible bias the sampling technique. This is actually possible for any studied lineage. The lack of detailed picking and sieving will affect the diversity. This is especially sensitive in the case of historical references, which often described material recovered from the surface instead of being the result of systematic excavations. Moreover, natural conditions can work in the same way. Sedimentary conditions can cause the lack of several forms by size; the environment biases at the moment of the burial can favor the deposit and abundance of some forms over others (for example, resistance to transportation can erase delicate lamniform teeth and will differentially affect small, compact teeth). These commented biases (and other relevant that should be considered) are not clearly treated in the text, and I think they are critical for the robustness of the results. The scale of such analysis pointing to a global interpretation is certainly enormous, reason why I would (again) recommend the authors to attempt the same analysis in localities with good sampling, with good control of the different drivers, and having enough control of the biases. Finally, if authors agree to consider these recommendations, I would additionally recommend to provide your own database, drivers references, and bias controls as supplementary data of the paper. On the contrary, you are forcing the reader to search the cited database(s) in case of having questions or criticisms, and finally, you are transferring the responsibility of any data incompleteness to an external source.

 Finally, I'm attaching an annotated pdf with minor recommendations. My most relevant comments are already here.

Author Response

Comment 1:

I must begin saying that complex numerical modellings were challenging during this current review. I do have concerns with relaying entirely on these type of modellings for assuring global hypotheses.

Response 1:

We would like to express our sincere gratitude for the constructive feedback provided by reviewer one, with whom we are in agreement regarding the processes at local scales in the context of global scale diversity. Notwithstanding, the present study constitutes a significant foundation for the exploration of large-scale/global trends and patterns, and such studies hold particular significance when comparing processes at higher taxonomic ranks or ecological groups. Consequently, our study seeks to establish a foundational framework upon which subsequent research can be built, with the aim of investigating smaller-scale processes in greater detail while incorporating this local information to enhance our global understanding.

This type of study, employing a larger-scale approach, has the potential to identify areas where we may be able to improve our sampling efforts, thus enhancing the quality of the available data. While focusing on single localities alone may not fully capture the gaps in palaeobiological research in terms of neoselachians, we believe that our approach offers a valuable contribution to the wider discussion.

Comment  2:

Databases should be considered as an incomplete approaching instead of confident sources of data. This because many localities around the planet are undersampled and all of them has biases that will affect the results.

Response 2:

We have chosen to adopt an integrative approach, which we believe has the potential to enhance the quality of the raw dataset. This approach involves the utilisation of rarefaction to ensure data integrity, sampling standardisation via shareholder quorum subsampling to account for potential sampling biases, and the employment of multiple integrative diversity measures to balance for the limitations of individual approaches. We consider the database to be a valuable resource, as it contains a substantial amount of fossil data, thereby constituting a suitable representation of global diversity, especially in combination with the aforementioned rarefaction analyses and sampling standardisation. However, it should be noted that results obtained from modelling should be considered as only an approximation to real-world phenomena, offering suggestions for potential relationships that allow for the establishment of hypotheses. Consequently, the results of our study are open to discussion and further research by the scientific community. We added a respective paragraph in the manuscript in the conclusions section.

Comment  3:

I do think that the structure of the manuscript and the carried out analysis is very valuable, but I would recommend to restrict it to localities with good sampling (i.e., large number of pieces) as a first goal, instead of pointing to global conclusions (which might be precipitate at this moment).

Response 3:

While we recognise the value of the data from individual localities (alpha diversity), it is important to acknowledge that this data often contains only data from a short period and includes only a relatively modest number of data points. This can make subsampling difficult, if not impossible in some cases. When considering only individual localities, they tend to be even more prone to incomplete sampling, as crucial parts of the locality may not have been investigated. Additionally, most localities often include just one (rarely more) habitat type(s), which can make discussions on patterns at higher taxonomic ranks or on different habitat preferences difficult. In our opinion, analyses of global/large-scale diversity (gamma-diversity) are a valid method, and they have been used frequently in the past by various authors (Alroy et al. 2008, Friedman & Sallan 2012, Guinot & Cavin_2016, Condamine et al. 2019, Guinot & Cavin_2020, Guinot & Condamine 2023, …). In this study, the chosen boundary crosser and range through diversity approach, for example, accounts for singletons.

Comment   4:

In general, several sections (e.g., Environmental Diversification Drivers, Biotic Drivers) deal easily with planetary variables (sea level, sea surface temperature, atmospheric CO2 concentration , flooded continental áreas, dinoflagellate, nannoplankton, planktic foraminifera, etc) which in many cases are far from being accurately controlled in several localities with relevant chondrichthyan records.

In addition, I really miss pivotal references on these sections, because each parameter has many previous studies and they are not reflected in the text.

Response 4:

References for the used environmental parameters are included in the material and methods section. The suggested inclusion of the respective references in the results section was adjusted by referring to the material and methods section.

Comment  5:

Again, my recommendation it that authors should build a solid case only restricted to those localities where these drivers were enough studied previously (adding to the previous recommendation of localities with good sampling numbers). Else, they risk to have artifactual results.

Response 5:

Please refer to the replies to previous comments. Further, a significant field of palaeo-statistics focuses on addressing the variations within the fossil record (see, for instance, the works of Alroy et al. and Foote et al.). This field has developed advanced methodologies to address these variations, including classical rarefaction and shareholder quorum subsampling. Nonetheless, it is imperative to recognise that the fossil record is inherently uneven and incomplete. Despite these limitations, the fossil record serves as the foundation for our research, necessitating its utilisation in our analytical studies. The fossil record of neoselachians is frequently regarded as the most complete of all vertebrates (although it remains a limited insight into the past), and thus can be regarded as one of the more reliable fossil records.

Comment  6:

Regarding the Conclusions, these reinforce previous hypotheses on the Jurassic and Cretaceous radiation events, which is great. Regarding the lamniform dominance starting the Late Cretaceous, the authors point as a possible bias the sampling technique. This is actually possible for any studied lineage. The lack of detailed picking and sieving will affect the diversity. This is especially sensitive in the case of historical references, which often described material recovered from the surface instead of being the result of systematic excavations. Moreover, natural conditions can work in the same way. Sedimentary conditions can cause the lack of several forms by size; the environment biases at the moment of the burial can favor the deposit and abundance of some forms over others (for example, resistance to transportation can erase delicate lamniform teeth and will differentially affect small, compact teeth).

Response 6:

When considering the preservation of fossils, in this case neoselachian remains (which are mostly teeth or scales), a number of factors must be taken into account. Nevertheless, the prevalence of large-toothed taxa within the neoselachian fossil record (predominantly Lamniforms) is evident. Consequently, we propose a probable size-mediated sampling bias (a suggestion that has been previously reported in Underwood 2006, Jambura et al. 2024, etc.). It is acknowledged that teeth and scales of smaller size and greater fragility may be more susceptible to the effects of drag, kinetic forces and associated processes. However, this results in the observed predominance of lamniform occurrences within the overall fossil records. Undoubtedly, intensified sampling efforts, especially for smaller taxa, would support the relative proportion of these groups. Therefore, we highly advocate focusing on fair sampling across all sizes in the field. We included taphonomic biases in our lamniform occurrence dominance discussion.

Comment  7:

These commented biases (and other relevant that should be considered) are not clearly treated in the text, and I think they are critical for the robustness of the results.

Response 7:

Taphonomic biases pose a substantial challenge when considered within the framework of large-scale studies, given the unique influence of each locality's distinct set of regional processes and chemical conditions.  In order to mitigate the influence of taphonomic biases to some degree, the application of sampling standardisation is recommended once again.

We would like to ask reviewer 1 to offer further suggestions with regard to which additional relevant factors, apart from those mentioned above, should be taken into consideration.

Comment  8:

The scale of such analysis pointing to a global interpretation is certainly enormous, reason why I would (again) recommend the authors to attempt the same analysis in localities with good sampling, with good control of the different drivers, and having enough control of the biases.

Response 8:

For many localities worldwide, the quality of sampling has never been analysed individually in detail. As long as the quality of the sampling of individual localities is not clarified, it is difficult to contextualise them and draw conclusions. However, we absolutely agree that such a study would be extremely valuable and highly desirable.

As already mentioned, our global study is intended to provide a basic framework on which further research can build to investigate processes on a smaller scale. It serves as an overview of global trends. Regional faunas also tend to reflect a very incomplete part of the diversity at a given time due to the incompleteness of the fossil record. When analysing supra-regional to global faunas of a specific age, it is possible to at least partially compensate for the fragmentary nature of small localities.  As also mentioned above, small-scale diversity analyses usually do not allow analyses in deep time, which in turn does not show changes over longer periods of time and thus does not allow comparisons with changing environmental conditions.

Comment  9:

Finally, if authors agree to consider these recommendations, I would additionally recommend to provide your own database, drivers references, and bias controls as supplementary data of the paper. On the contrary, you are forcing the reader to search the cited database(s) in case of having questions or criticisms, and finally, you are transferring the responsibility of any data incompleteness to an external source.

Response 9:

Our database is of course available on reasonable request. The utilised sources of the data for the environmental drivers are cited in the materials and methods section.  In addition, the first and last appearance date for the individual genera can be found in the supplementary file, which should enable the performance of all diversity analyses. The entire occurrence dataset will eventually be made available on request and via a data repository. Currently, however, this database forms the basis of an ongoing PhD project, and it is anticipated that it will be further utilised in the very near future by that project.

Comment  10:

 Finally, I'm attaching an annotated pdf with minor recommendations. My most relevant comments are already here.

              Comment 10A:

                            Response 10A: Adjusted accordingly

              Comment 10B:

                            Response 10B: Adjusted accordingly

              Comment 10C:

                            Response 10C: Adjusted accordingly

              Comment 10D:

                            Response 10D: Adjusted accordingly

Reviewer 2 Report

Comments and Suggestions for Authors

I have gone through the manuscript, and it is found very interesting to me. I would suggest it to get published in your journal. The authors discussed the role of atmospheric CO2 and tectonic in the diversification of elasmobranchs specially of the Mesozoic time.  Even though I personal feel that certain additional factors may also contribute to this beside the concentration of CO2. Still, I much appreciate the quantum of data they analysis, discussed and interpreted. The events of evolution, diversification, extinction are complex and multidimensional in nature specially in the case of more advanced fauna. However, the authors could demonstrate the negative effect of higher atmospheric CO2 concentration on neoselachian diversity in the past which is quite appreciable and need to be highlighted. I suggest the manuscript to be accepted for publication after taking into consideration of minor typological and grammatical mistakes. 

Comments on the Quality of English Language

Minor grammatical and typological errors may be rectified before publication.

Author Response

Comment  1:

Minor grammatical and typological errors may be rectified before publication.

Response 1:

We would like to thank reviewer two for suggesting improvements to the grammar and spelling, which were very much appreciated. The manuscript was reviewed very thoroughly, and it is worth noting that the involvement of a native English speaker (MJB) has contributed to enhance the manuscript's quality in this regard.

Round 2

Reviewer 1 Report

Comments and Suggestions for Authors

Congratulations to the authors for the substancial effort in attending the comments made on round 1. The manuscript reads easily, and the needed references are oportune on each section. The additions to the Discussion are also much welcome, same the robustness of the new version of Supplementary Data. The conclusions are better arranged and each discussed topic is well reflected in the former.

Said that, I don't have further comments and in my opinion, the manuscript is fine for publication.